# Reassessment of Viroid RNA Cytosine Methylation Status at the Single Nucleotide Level

**DOI:** 10.3390/v11040357

**Published:** 2019-04-18

**Authors:** Francesco Di Serio, Enza Maria Torchetti, José-Antonio Daròs, Beatriz Navarro

**Affiliations:** 1Istituto per la Protezione Sostenibile delle Piante (CNR), 70126 Bari, Italy; francesco.diserio@ipsp.cnr.it (F.D.S.); enza.torchetti@gmail.com (E.M.T.); 2Instituto de Biología Molecular y Celular de Plantas (Consejo Superior de Investigaciones Científicas-Universitat Politècnica de València), 46022 Valencia, Spain

**Keywords:** viroid RNA, bisulfite sequencing, nucleotide modification, C5-methylcytosine

## Abstract

Composed of a few hundreds of nucleotides, viroids are infectious, circular, non-protein coding RNAs able to usurp plant cellular enzymes and molecular machineries to replicate and move in their hosts. Several secondary and tertiary RNA structural motifs have been implicated in the viroid infectious cycle, but whether modified nucleotides, such as 5C-methylcytosine (m^5^C), also play a role has not been deeply investigated so far. Here, the possible existence of m^5^C in both RNA polarity strands of potato spindle tuber viroid and avocado sunblotch viroid -which are representative members of the nucleus- and chloroplast-replicating viroids, respectively- has been assessed at single nucleotide level. We show that a standard bisulfite protocol efficiently used for identifying m^5^C in cellular RNAs may generate false positive results in the case of the highly structured viroid RNAs. Applying a bisulfite conversion protocol specifically adapted to RNAs with high secondary structure, no m^5^C was identified in both polarity strands of both viroids, indicating that this specific nucleotide modification does not likely play a role in viroid biology.

## 1. Introduction

One of the most surprising and fascinating traits emerging in the last decades in molecular biology is the central role of RNA as regulatory molecule in many and very different biological processes [1]. Primary, secondary and tertiary structural elements have been specifically involved in RNA functionality. In this context, modified nucleotides have been reported providing functional roles to several cellular RNAs, including ribosomal RNAs (rRNAs), transfer RNAs (tRNAs), messenger RNAs (mRNAs), small nuclear RNAs (snRNAs), long non-coding RNAs (lncRNAs) and others [2,3]. Implication of RNA methylation in the initiation and progression of cancer [4] and in the prevention of recognition of invasive RNAs by the innate human immune system [5] have also been documented. Methylation, which occurs in a dynamic manner [6], is one of the most common nucleotide modifications in RNA. Identification of methylated bases in RNA, including C5-methylcytosine (m^5^C) and N6-methyladenosine (m^6^A), dates back to about 50 years ago [7,8]. Initially, nucleotide modifications were thought to be mainly restricted to the stable and highly abundant tRNAs and rRNAs. In these cellular RNAs, the modified nucleotides, including m^5^C, were mapped in technically demanding studies that support their possible structural and/or functional roles [9,10,11,12]. More recently, the employment of immunoprecipitation combined with deep sequencing allowed to gain a transcriptome-wide picture of m^6^A and m^5^C, unveiling the relevance of this nucleotide modification in mRNAs [13,14,15,16]. Similarly, the improvement of bisulfite sequencing protocols for RNA [17,18], originally developed for identifying m^5^C in DNA, allowed to overcome the main limitation to the extension of this technique to RNA analysis, thus unveiling the widespread occurrence of m^5^C in cellular coding and non-coding RNAs [19,20]. Although functional roles of RNA methylation are still unclear, these findings showed that the previous reports on the limited presence of methylated nucleotides in cellular RNAs other than rRNA and tRNAs were mainly due to technical limitation [2]. The current availability of innovative approaches for RNA methylation analysis completely changed the general picture on the presence and possible functions of modified nucleotides in cellular RNAs [21].

Viroids are non-coding RNAs that infect higher plants [22,23,24]. With a tiny, circular, single-stranded RNA genome, viroids are able to interact directly with host-encoded molecular factors needed for their replication and systemic movement. This property is likely due to the viroid ability to mimic structural features of host RNAs, so that the infectious RNAs can usurp cellular enzymes and molecular machineries and redirect them to their own replication and spread in the infected host [25]. In agreement to this prediction, several structural elements of viroid RNAs have been shown to play major roles in the infectious cycle [26], including tertiary structural motifs that are involved in replication and/or movement of viroid RNAs [27]. Viroids from different families replicate through alternative rolling circle mechanisms (symmetric and asymmetric), in which host DNA-dependent RNA polymerases use viroid RNAs as templates to synthesize complementary RNA molecules. By convention, + polarity is assigned to the viroid strand that accumulates at higher level in the infected tissues. According to the replication sites, the rolling circle replication mechanism and other structural and biological features, viroids are grouped in two families: *Pospiviroidae*, which groups nuclear viroids replicating though an asymmetric rolling circle mechanism, and *Avsunviroidae*, including the chloroplast viroids that replicate through a symmetric rolling circle mechanism [28,29].

Information on whether viroid RNAs, similarly to other highly-structured cellular RNAs such as tRNAs or rRNAs, contain modified nucleotides is currently very limited. To our knowledge, the only attempt to explore this dates back approximately four decades ago to early studies on the primary and secondary structures of potato spindle tuber viroid (PSTVd) molecule [30]. Based on the notion that deamination of cytidine to uridine induced by bisulfite treatment depends on the secondary structure of the RNA -cytidine is fully reactive in single-stranded RNAs, while it is resistant against modification in base-paired structures [31,32]- RNase fingerprints analyses of bisulfite-treated PSTVd RNA preparations allowed Domdey and colleagues to conclude that the reactivity of this viroid against bisulfite-catalyzed modification of cytidines to uridines was consistent with a rod-like secondary structure composed of short unpaired loop regions connected by short helices [30]. Moreover, modified nucleotides at the 5′-end group of the oligonucleotides generated by RNase digestions were not detected in PSTVd RNA preparations tested by fingerprints, thus suggesting the lack of m^5^C at these positions in the viroid RNA. This study was limited to the mature circular PSTVd RNA of + polarity. Moreover, the approach followed by Domdey et al. was unable to detect m^5^C at a single nucleotide resolution or in a minor fraction of potentially methylated RNAs. For this reason, in this work, using updated techniques, we aimed to re-investigate whether viroid RNAs are methylated. First, we adapted a specific protocol based on RNA bisulfite sequencing to reassess, at a single nucleotide resolution level, the question regarding the presence of m^5^C in viroid RNAs. We applied this protocol to the strands of both polarities of PSTVd and avocado sunblotch viroid (ASBVd), type species of the families *Pospiviroidae* and *Avsunviroidae,* respectively, isolated from infected host plants. Remarkably, experimental data confirmed the absence of m^5^C in both strands of both viroids. Moreover, comparison of this protocol and a previously established protocol to study m^5^C in mRNAs [18] showed that ours is more appropriate to analyze the methylation status of viroid and possibly other highly structured RNAs, which are particularly resistant to bisulfite conversion due to the high level of base-paired cytidines.

## 2. Materials and Methods

### 2.1. Plant Material and RNA Preparations

Leaf samples (10 g) were collected from *Nicotiana benthamiana* and avocado (*Persea americana*) plants infected with PSTVd and ASBVd, respectively, and total nucleic acid were extracted with phenol and enriched in highly structured RNAs, including also tRNAs and viroid RNAs, by a chromatography on CF11 cellulose as reported previously [33].

### 2.2. Bisulfite Sequencing Protocols

Aliquots (3 µg) of viroid enriched RNA preparations were treated with 2 U of RNase-free DNase (RQ1; Promega, Madison, WI, USA) following the supplier’s protocol. After phenol extraction and ethanol precipitation, RNA was resuspended in 10 µL of water to perform bisulfite treatment according to the protocol reported by Pollex et al. [18], hereafter defined as standard (St) protocol. Alternatively, a modified protocol, herein defined improved (Im), was applied. When the St protocol was used, 42.5 µL of bisulfite mix and 17.5 µL of DNA protect buffer (EpiTect Bisulfite kit, Qiagen, Hilden, Germany) were added and the resultant mixture was incubated in a thermal cycler to run the following program: four cycles of 5 min at 70 °C, followed by 60 min at 60 °C. When the Im protocol was performed, the incubation in the thermal cycler was as follows: one cycle of 5 min at 80 °C, followed by 60 min at 60 °C, plus four cycles of 5 min at 75 °C, followed by 60 min at 60 °C. The reaction mix was then desalted by size-exclusion chromatography using a Micro Bio-Spin 6 column (Bio-Rad, Hercules, CA, USA) and the RNA adducts desulfonated by adding 1 volume of 1 M Tris-HCl (pH 9.0) and incubating at 37 °C for 1 h. Converted RNA was precipitated by adding sodium acetate (pH 5.2) to 0.3 M, 1 µg of glycogen, and 3 volumes of 96% ethanol.

### 2.3. RT-PCR, Cloning and Sequencing

An aliquot (2 µL) of bisulfite converted RNA was used for performing the reverse transcription with random primers and M-MuLV Reverse Transcriptase (New England BioLabs; Ipswich, UK). cDNAs were amplified by PCR using Taq DNA polymerase and the Expand High Fidelity PCR system (Roche Applied Science, Penzberg, Germany) in combination with primers appropriately designed to minimize cytosine content (Table 1). The amplification products were electrophoresed in a 5% agarose gel, eluted, ligated in the pGem-T-easy vector (Promega, Madison, WI, USA) according to the manufacturer protocol, cloned and sequenced with an ABI 3100 XL (Life Technologies, Camarillo, CA, USA) apparatus.

## 3. Results and Discussion

Bisulfite sequencing protocols are generally based on a first step in which RNA preparations are treated with sodium bisulfite to convert non-methylated Cs to Us, and on a second step in which the treated RNAs are reverse transcribed, cloned and sequenced to detect converted and unconverted Cs, likely corresponding to unmethylated and methylated residues, respectively, in the original RNA [17]. In this work, appropriate controls were selected to test the conversion efficiency of non-methylated Cs to Us by the bisulfite treatment. To this aim, on the one hand, the tRNA-Asp was selected as an internal control. It is well known that the C at position 38 (C38) of this RNA is methylated, in contrast to Cs in close positions that are unmethylated [34]. On the other hand, since C-G base-paring in self-complementary RNA molecules may impair the efficiency of bisulfite conversion [31,32], an additional control to explore whether lack of bisulfite conversion of some Cs is due to base-pairing instead of the actual nucleotide methylation was also included. Although RNA modifications in vivo may further stabilize the RNA secondary structure, we reasoned that a full-length RNA transcript of hop stunt viroid (HSVd) generated in vitro -which is not methylated- is actually expected to be fully converted by bisulfite treatment. Unconverted Cs detected in this transcript would suggest an inefficient treatment or a resistance to the bisulfite conversion, possibly due to local base pairing in the RNA. HSVd was selected to this aim because (i) it assumes a rod-like conformation similar to those proposed for PSTVd and ASBVd [35], thus providing a proper control for testing the resistance to bisulfite conversion caused by inefficient RNA denaturation of highly structured viroid RNAs; and (ii) its size is similar to those of PSTVd and ASBVd.

To amplify cDNAs from RNAs with converted or unconverted Cs with minimum bias, it is critical to design the primers in sequence regions lacking Cs. However, this is particularly difficult in the case of most viroids, which are GC-rich [36]. Therefore, specific primers targeting viroid RNAs containing no or at the most one C were designed. In this case, the position corresponding to the C was degenerated to C/T and G/A in the forward and the reverse primers, respectively. Based on this strategy, two sets of primers were designed to investigate the methylation of the PSTVd RNA of + polarity, allowing the study of 97% of Cs. In contrast, a single set of primer was designed in the case of the − polarity strand of PSTVd, both + and − polarities of ASBVd, and + polarity of HSVd RNA, thus allowing the study of 90, 95, 94 and 79% of Cs contained in the genome of these viroid RNAs, respectively (Table 1).

When the protocol St was applied, the unconverted C at position 38 of the tRNA-Asp was detected in all the sequenced clones of *N. benthamiana*, whereas the proximal Cs were completely converted to Us, thus suggesting that the protocol is efficient for testing m^5^C in a tRNA. When the same protocol was applied to test PSTVd and ASBVd RNAs, several Cs remained unconverted in some of the tested clones. More specifically, 0.92% and 6.77% of Cs were not converted in a total of 39 and 21 cDNA clones of the + and − PSTVd strands, respectively (Figure 1, Figure 2 and Appendix A). Although at a lower extent, unconverted Cs were also detected in both polarity strands of ASBVd (0.30 and 0.93% in the + and − polarity, respectively) (Figure 1 and Appendix A).

Interestingly, unconverted Cs, although with a different frequency (up to 6.77% of total Cs), were identified in the clones obtained with all primer sets (Figure 2 and Appendix A). Surprisingly, one unconverted C was also found in one cDNA clone of the HSVd transcript RNA, where no methylation was expected (Appendix A). This finding, together with the lack of positions consistently methylated in the viroid RNAs purified from infected plants, suggested that the resistance to conversion could not reflect the actual nucleotide methylation status, but being a consequence of strong base pairing.

This hypothesis was further tested by repeating the experiment with the Im protocol that, adopting more exhaustive denaturing conditions during the bisulfite treatment, was expected to improve the conversion of Cs in highly structured RNAs. When such a protocol was applied, the methylation status of Cs in both the tRNA-Asp and HSVd transcripts was as expected, with methylated C38 in the tRNA and absence of methylated Cs in the in vitro viroid RNA transcript confirmed in all the cloned cDNAs from bisulfite treated RNA preparations. No unconverted C was detected in the cDNAs of both polarity strands of ASBVd by the Im protocol (Appendix A). Moreover, the unconverted Cs in the clones of PSTVd cDNAs dropped down to 0.08 and 0.07% for the + and − polarity strands, respectively (Figure 1, Figure 2 and Appendix A). Importantly, in the case of + PSTVd polarity, the single unconverted C identified with Im protocol in a single clone was in a position different from those detected with the St protocol. These results show that the Im protocol is better to avoid false positive results on the presence of m^5^C in the case of highly structured RNAs like viroids.

Altogether, these results indicate that the Cs of PSTVd and ASBVd are likely unmethylated in both polarity strands. However, it is important to highlight that this hypothesis could be further tested by coupling the effective bisulfite conversion protocol developed here with a high-throughput sequencing–based analysis of the bisulfite converted RNAs. This approach would explore in more exhaustive manner the possible presence of minor populations of viroid molecules with m^5^C. Alternatively, immunoprecipitation of methylated RNA with a specific antibody coupled with high-throughput sequencing could also be useful to this aim [37,38].

On the one hand, our data confirm the previous results on the + polarity strand of PSTVd reported early after viroid discovery by a less sensitive technique [30]. On the other hand, this is the first time that RNA methylation is investigated in the case of ASBVd, a completely different viroid that replicates in the chloroplast and belongs to a different family (*Avsunviroidae*). Our study shows that nuclear and chloroplast-replicating viroids do not largely differ from each other in respect to methylation and substantially diminish the possibility that m^5^C may have a role in viroid biology. In fact, the absence of methylation in both polarity strands support the conclusion that such a nucleotide modification is very likely not involved in viroid intracellular trafficking. Methylation of m^5^C has also been involved in the stability of some RNAs [39,40]. However, based on our results, this does not apply to both nuclear and chloroplast replicating viroids that have been shown to accumulate in vivo as free nucleic acids [41,42], whose stability is likely provided by their circularity and high degree of secondary structure. Finally, since it has proposed that RNA methyltransferases appeared during the transition of RNA world to the present cellular life based on RNA/DNA/protein world [43], the question arises whether the absence of m^5^C in viroid RNAs could be considered as an additional indirect evidence of their ancient origin in the RNA world, besides their small, circular non-protein coding genomes, sometimes endowed of ribozyme activity [44]. Importantly, our study does not rule out the possibility of other nucleotide modifications in viroid biology, what remains a fascinating topic waiting for further research.

## Figures and Tables

**Figure 1 viruses-11-00357-f001:**
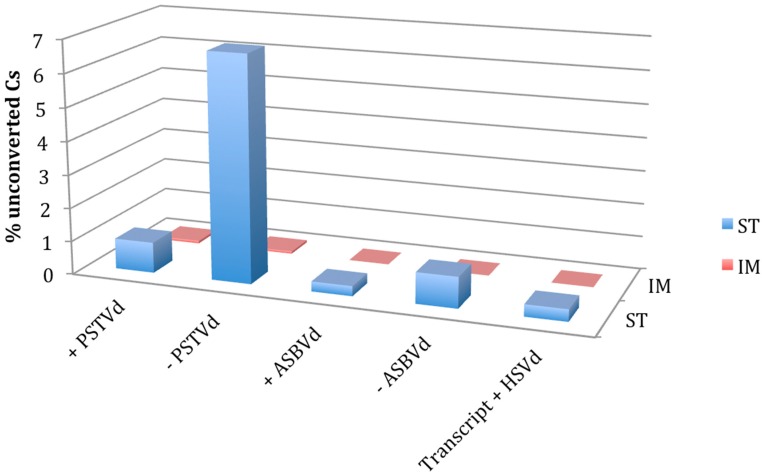
Comparison of RNA bisulfite sequencing by standard (St) and improved (Im) protocols. Percentage of unconverted cytosines detected in + and − strands of potato spindle tuber viroid (PSTVd) and avocado sunblotch viroid (ASBVd) infecting their natural hosts and in the in vitro transcript of the + strand of hop stunt viroid (HSVd) used as a negative control.

**Figure 2 viruses-11-00357-f002:**
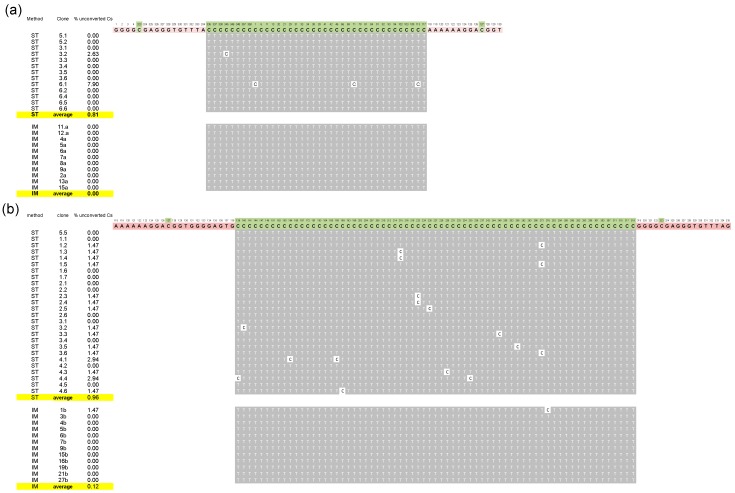
Bisulfite sequencing of + PSTVd RNAs. Bisulfite cytosine conversion generated by the standard (St) and the improved (Im) protocols of two + PSTVd fragments amplified with the primer pairs PSTVd_met_1F_plus/PSTVd_met_2R_plus and PSTVd_met_3F_plus/PSTVd_met_4R_plus (panel (**a**,**b**), respectively). The sequences targeted by the specific primers (salmon background) and the positions of cytosines in the reference variant PSTVd-Nb (green background) are indicated on the top. Converted (T, grey background) and unconverted (C, white background) cytosines are reported for each sequenced clone, the name of which is reported on the left, together with the percentage of unconverted cytosines.

**Table 1 viruses-11-00357-t001:** Primers used in this study.

Name	seq (5′ to 3′) *	Position ^#^
PSTVd_met_1F_plus	GGGGCGAGGGTGTTTAG	319–335
PSTVd_met_2R_plus	CACTCCCCACC**R**TCCTTTTTT	138–118
PSTVd_met_3F_plus	AAAAAAGGA**Y**GGTGGGGAGTG	118–138
PSTVd_met_4R_plus	CTAAACACCCTC**R**CCCC	335–319
PSTVd_met_5F_minus	GAAGAAAGGAAGGGTGAAAA	196–177
PSTVd_met_6R_minus	ACCACCCCTC**R**CCCCCTT	222–239
ASBVd_met_1F_plus	GTGGTGAA**Y**TTTTATTAAAAAAATTAG	106–132
ASBVd_met_2R_plus	CCAC**R**ACTCCTCCTTCTCTCACAA	109–86
ASBVd_met_3F_minus	GAGTGAA**Y**TAATTTTTTTAATAAAAGTT	139–112
ASBVd_met_4R_minus	TCTTCAATCTCTTRATCACTTC	141–162
HSVd_met_1F_plus	GAGAGGYGTGGAGAGAGGG	106–125
HSVd_met_2R_plus	CCTCCCTRCCTTATTTTTTCTTT	56–34
tRNA-Asp_1F	GTCGTTGTAGTATAGTGG	
tRNA-Asp_2R	ATCGTTCCCAGGTCAGGG	

* R is G or A; Y is C or T; ^#^ nucleotide positions related to plus strand of PSTVd variant Nb (AJ634596.1), ASBVd reference; variant (NC_001410.1) and HSVd reference variant (NC_001351.1).

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
