# Peer review of "Reassessment of Viroid RNA Cytosine Methylation Status at the Single Nucleotide Level"

_viruses, 2019, doi:10.3390/v11040357_

Reviewer 1 Report

In this manuscript Di Serio et al analyze the presence of cytosine methylation on viroid RNAs with a single nucleotide resolution. Because viroids are models of RNA biology it is very interesting to understand if RNA postranscriptional modifications can influence any aspect of their life cycle. Interestingly, Di Serio and colleagues find that cytosine methylation is found at very low levels in any of the strands of the two "model" pospiviroidae and avsunviroidae members: PSTVd and ASBVd. The manuscript is really well written and the experiments are well design. Although their results are negative it is in fact a confirmation of the lack of importance of this modification for viroid biology. My only comment is that in the conclusion paragraph it could be interesting to comment that an alternative technique to enrich potential viroidal RNAs with high levels of m5C could be performing immunoprecipitations with a m5C antibody. Other than that I consider that the manuscript could be accepted in its present form.

Minor comments:

-Line 185: Typo: "in very likely" should be "is very likely"

-Line 187: Typo "an high-throughput" should be "a high-throughput"

-Lines 178-180: "On the other hand.." is used in two consecutive sentences please use a different connector.

Author Response

Answers (A) to reviewer’s comments (C):

Reviewer 1:

C: My only comment is that in the conclusion paragraph it could be interesting to comment that an alternative technique to enrich potential viroidal RNAs with high levels of m5C could be performing immunoprecipitations with a m5C antibody

A: We agree with the reviewer that m5C immunoprecipitation techniques must be highlighted. We added a specific sentence in the conclusion paragraph (see L 228-229) and mentioned the techniques more explicitly in the introduction (L 39). We also quoted new pertinent references (see References 15, 16).

 C: -Line 185: Typo: "in very likely" should be "is very likely"

A: Done

 C: -Line 187: Typo "an high-throughput" should be "a high-throughput"

A: Done

 C: -Lines 178-180: "On the other hand.." is used in two consecutive sentences please use a different connector.

A: We used in the first sentence “On the one hand” and in the second one “One the other hand” (see L 230-231).

Reviewer 2 Report

Authors analysed the presence/absence of Cs methylation in two viroids genomes following 2 protocols, the second implemented by the authors. The second protocol validates the absence of methylation in the two viroids suggesting that Cs methylation is not involved in this two viroids biology

Author Response

The reviewer has no comments to be answered

Reviewer 3 Report

In the manuscript, entitled “Reassessment of viroid RNA cytosine methylation status at the single nucleotide level” the authors describe a slightly modified bisulfite conversion protocol used to study m5C levels in viroid RNA. They confirm the results of an old study (with less sensitive techniques) on PSTVd and show for the first time the lack of cytosine methylation on Avsunviroidae

The manuscript is clear and well written.

In my opinion the authors try to oversell the improved bisulfite method. The so-called “improved method” only differs slightly, and other groups, in other fields, have published other more stringent bisulfite treatments.  Nevertheless, it is an improvement from the actual standard protocol.

It really is a pity that the authors did not use NGS, but Sanger sequencing. However, I do realise it is too much to ask for NGS instead of Sanger, since this would mean having to repeat all experiments. However, not only could this have saved the authors a lot of time, but it would also have given much more data. The authors have given a lot of thought on primer design and appropriate controls. Yet, using for example the TruSeq Small RNA library prep kit (Illumina) or similar kits from other companies, would have given much more data and no need for primer design. For example, for control tRNA-Asp (line 146) it is stated that “at least three clones for each treatment” were sequenced. Only three.

Concerning appropriate controls: since in vivo modifications might stabilize the RNA structure, Cs in the in vitro HSVd construct might still be easier accessible due to less (strong) secondary structures.  Please comment in the text.

Line 154 most unconverted Cs were found only in one clone – not special, often the case (as seen in high throughput bisulfite sequencing).

Line 122 tRNA-Asp from what? Both Nicotiana benthamiana and avocado plants? Please clarify in the text.

In conclusion: I do believe that the experiments are carefully performed with attention for detail. However, the technique is old fashioned, yet not incorrect and still reliable. Similar (not identical) improvements to the bisulfite conversion have been shown in other fields.

I would recommend the authors to focus more on the viroid part, rather than the technical part. So please shorten the technical part. Maybe the authors could think about why C methylation is absent? Both from evolutionary as from functional perspective (what would be the difference in how it influences the hosts machinery?

Author Response

Answers (A) to reviewer’s comments (C):

Reviewer 3

C: In my opinion the authors try to oversell the improved bisulfite method. The so-called “improved method” only differs slightly, and other groups, in other fields, have published other more stringent bisulfite treatments.  Nevertheless, it is an improvement from the actual standard protocol

A: We agree with the reviewer that other protocols have been published to increase the stringency of RNA bisulfite conversion in other fields. Actually, one of the aims of this work was to extend the use of this technique to viroid RNAs, which are quite peculiar RNAs because of their high secondary structure and circularity. In this respect, we think our data provide useful indications to increase reliability of bisulfite sequencing of viroid RNAs. We modified the text in the introduction to better clarify this point (L 84 and L 90-91). We appreciate that the reviewer recognized that our manuscript provides an improvement in this direction.

C: It really is a pity that the authors did not use NGS, but Sanger sequencing. However, I do realise it is too much to ask for NGS instead of Sanger, since this would mean having to repeat all experiments. However, not only could this have saved the authors a lot of time, but it would also have given much more data. The authors have given a lot of thought on primer design and appropriate controls. Yet, using for example the TruSeq Small RNA library prep kit (Illumina) or similar kits from other companies, would have given much more data and no need for primer design.

A: As commented in the manuscript, we understand that NGS would provide additional and possibly more conclusive data. However, as also reported in the answer to the previous comment, this is the first attempt of using bisulfite sequencing to explore the methylation in viroid RNAs and we provided evidence that the conversion can be more effective using the adapted protocol. Therefore, data based on Sanger pave the way for further high-throughput analyses that can be performed by us or by other research groups.

C: For example, for control tRNA-Asp (line 146) it is stated that “at least three clones for each treatment” were sequenced. Only three.

A: Since we obtained always 100% conversion of tRNA with the standard protocol, we considered this control less relevant to our task of testing methylation in the larger and highly structured viroid RNAs. For this reason, we used the additional control based on the in vitro HSVd transcripts in our experiments.

C: Concerning appropriate controls: since in vivo modifications might stabilize the RNA structure, Cs in the in vitro HSVd construct might still be easier accessible due to less (strong) secondary structures.  Please comment in the text.

A: We agree. A specific comment has been done in this respect (L 151-154.): “Although RNA modifications in vivo may further stabilize the RNA secondary structure, we reasoned that a full-length RNA transcript of hop stunt viroid (HSVd) generated in vitro.... is actually expected to be fully converted by bisulfite treatment”.

C: Line 154 most unconverted Cs were found only in one clone – not special, often the case (as seen in high throughput bisulfite sequencing).

A: The sentence has been removed

C:Line 122 tRNA-Asp from what? Both Nicotiana benthamiana and avocado plants? Please clarify in the text.

A: We clarified in the text that the assay was done for N. benthamiana (L 171).

C: I would recommend the authors to focus more on the viroid part, rather than the technical part. So please shorten the technical part. Maybe the authors could think about why C methylation is absent? Both from evolutionary as from functional perspective (what would be the difference in how it influences the hosts machinery?

A: We thank very much the reviewer for the invitation to provide additional comments on the possible implications of our results on viroid biology from a functional and evolutionary point of view. In the last part of the manuscript, we added several considerations in this respect, thus extending the discussion regarding viroid biology and, at the same time, improving the balance between the biological and the technical part. If possible, we prefer not to shorten the technical part because, as commented previously, this is the first attempt of applying bisulfite sequencing to the study of viroid RNAs and technical details would be relevant for scientists interested in using the proposed protocol for further studies.